# Comprehensive Longitudinal Linear Mixed Modeling of CTCs Illuminates the Role of Trop2, EpCAM, and CD45 in CTC Clustering and Metastasis

**DOI:** 10.3390/cancers17162717

**Published:** 2025-08-21

**Authors:** Seth D. Merkley, Huining Kang, Ursa Brown-Glaberman, Dario Marchetti

**Affiliations:** 1Division of Molecular Medicine, Department of Internal Medicine, University of New Mexico Health Sciences Center, Albuquerque, NM 87131, USA; smerkley@salud.unm.edu; 2Division of Epidemiology, Biostatistics and Preventive Medicine, Department of Internal Medicine, University of New Mexico Health Sciences Center, Albuquerque, NM 87131, USA; hukang@salud.unm.edu; 3Christus St. Vincent Regional Cancer Center, Santa Fe, NM 85705, USA; ursa.brownglaberman@stvin.org

**Keywords:** EpCAM, Trop2, CTC, CTC clusters, breast cancer metastasis, liquid biopsy, Cytokeratin, CD45

## Abstract

Circulating tumor cells (CTCs) are the primary mediators of metastatic disease. After leaving the tumor, they can travel through the blood alone or with cluster partners to aid them on their way to distant metastatic sites. While CTCs are critical diagnostic tools for monitoring cancer progression, the biology underlying their behavior is obscured by their rarity and heterogeneity. To unravel the complexity of their actions, we undertook a large-scale longitudinal study to determine the factors that affect their presence, clustering, and their impacts on disease progression.

## 1. Introduction

Breast cancer is the most frequently diagnosed cancer in the world, and results in an estimated 685,000 deaths every year, with this burden projected to increase substantially over the next two decades [1,2]. While great strides have been made with increasingly potent and specific chemotherapeutics such as antibody–drug conjugates [3,4], treatment remains challenging because breast cancer is a heterogeneous disease spanning diverse subtypes, each with differing biomarker expression, prognoses, and metastatic capacities [5,6,7]. However, across breast cancer and all cancer types, 90% of deaths are due to distant metastasis from the primary tumor [8,9].

Breast cancers are classified based on expression of the estrogen receptor (ER), the progesterone receptor (PR), and human epidermal growth factor receptor2 (HER2). In particular, tumors are often grouped into three categories: hormone receptor positive in the absence of HER2 expression (HR+), HER2-expressing in the presence or absence of hormone receptor expression (HER2+), or negative for all three biomarkers (triple negative, or TNBC). HR+ cancers carry the best prognosis and are targeted by many well-established therapeutics [10,11]. While HER2+ cancers are more aggressive and carry a higher metastatic risk, the development of numerous highly effective anti-HER2 therapies has dramatically improved the prognosis of this subtype in the last two decades [12,13]. TNBC is the most aggressive and least-treatable subtype, with a 5-year mortality rate of 40%, a median survival time of 13.3 months after metastatic diagnosis, and a recurrence rate as high as 25% [3,14,15].

Regardless of subtype, metastasis spreads via circulating tumor cells (CTCs), which disseminate from the primary tumor through the vasculature to secondary sites where they enter dormancy and/or trigger distant metastasis. Breast cancer CTCs are classically defined as CD45–, EpCAM+, and Cytokeratin 8/18/19+ (classical CTCs or cCTCs), and this definition has yielded important insights into how CTC shedding predicts survivorship [16]. EpCAM is a transmembrane glycoprotein with homeostatic roles in the maintenance of epithelial barrier integrity, while elevated EpCAM expression within solid tumors is associated with poor prognosis [17]. EpCAM is a central regulator in many critical tumorigenic processes, including cell proliferation, adhesion, and migration [12,17,18], and has been explored as a druggable target in many solid tumors. For example, efficacy of CAR-T cell immunotherapy targeting EpCAM has recently been demonstrated [19].

Cytokeratins (CK) are intermediate filament proteins known to regulate structural integrity of epithelial cells. In addition to being markers for classical CTCs, CKs are also widely overexpressed within solid breast tumors and associated with poor survivorship [20,21]. Cytokeratins 8 and 18 (CK8 and CK18) are co-expressed in healthy mammary glands, and while their expression in tumor tissue negatively correlates with rates of recurrence, ER status, and tumor grade, CK8/18 are expressed by breast cancer CTCs [21]. CK19 has roles in cell-adhesion, motility, and maintenance of epithelial morphology, is linked to recurrence, and is also expressed by breast cancer CTCs [20,21,22,23]. Curiously, despite their shared presence on breast cancer CTCs, overexpression of CK18 and CK19 in tumor tissues correlates in opposite directions with disease progression, though this may differ by subtype [24]. Similarly to EpCAM, CK18 and CK19 are both reliable diagnostic markers in addition to being promising therapeutic targets [19,20,21].

Despite advances made using the classical CTC paradigm, it is also clear that many metastasis-competent CTCs do not fit the classical definition. For example, downregulation of EpCAM and Cytokeratins (CK) in CTCs is known to be associated with a mesenchymal subtype, as is upregulation of Vimentin [25,26]. While the epithelial-to-mesenchymal transition (EMT) is typically seen as a key step in metastatic progression, it is not a requirement for breast cancer lung metastasis. Within-tumor heterogeneity in EMT status may confer complementary benefits as non-EMT cells are metastatically competent and EMT cells can facilitate chemoresistance and recurrence at metastatic sites [27].

Furthermore, certain tumor-specific markers such as HER2 or EGFR may not be present in CTCs due to tumor heterogeneity, early dissemination, epigenetic changes, and/or drug-induced selection events [5,25,28]. Indeed, biomarker discordance between CTCs and the tumors they shed from has been frequently documented [25,28,29,30]. For these reasons, expansion of biomarker-based classifications, biomarker-agnostic approaches such as microfluidics based on size/deformability/density, and negative depletion of CD45+ immune cells are employed to cast a wider net in characterizing neoplastic cells outside of the classical CTC classification [25,31,32,33,34]. Recent research from our lab and others has highlighted EpCAM’s only homolog, Trop2 [7,35,36] (also known as EpCAM2), as a promising CTC biomarker and target in aggressive and otherwise untreatable HER2+ and TNBC diseases [3,4]. Trop2 shares many functions with EpCAM, including stabilization of tight-junction proteins and oncogenic roles in proliferation, adhesion, and migration [37,38]. However, they also undergo distinct post-translational modifications, show different patterns of expression in healthy and cancerous tissues [39,40], and while both allow for cell contractility, they may play discrete roles in fine-tuning adhesion and migration [41].

EpCAM and Trop2 promote homophilic cell adhesion in addition to their roles in stability of tight-junction Claudins in healthy epithelia and neoplastic cells [37,38]. Due to the relevance of these mechanisms to CTC motility, further longitudinal studies utilizing large datasets are necessary to unravel the underlying complexity of EpCAM’s role in CTC dissemination [17,18,40,42]. Indeed, some evidence suggests that despite EpCAM’s known roles as an adhesion molecule, surface expression can also impair adhesion [18].

CTCs in blood travel as single cells, clusters of homotypic CTCs, or in close association with CD45+ immune cells, myeloid-derived suppressor cells [43], platelets, or cancer-associated fibroblasts. Evidence suggests diverse benefits of clustering with stromal and immune cells, including immune evasion, anoikis resistance, resistance to shear- and oxidative stress, and 20–100× improved metastatic competency relative to single CTCs [6,44,45,46]. Additionally, non-tumor cells associated with CTCs may be a potent source of growth factors and cytokines, which eventually aid in the establishment of the distant pre-metastatic niche [44,45,47]. While EpCAM and Trop2 are relatively weak cell adhesion molecules compared to the classical junction proteins such as E-Cadherin [17], this may be an advantage for forming clusters with tumor-resident immune cells prior to invasion of the surrounding connective tissue. EpCAM and/or Trop2 expression may allow for transient but stable cell–cell contacts to be maintained in the absence of contact inhibition and polarization, allowing increased motility and migratory capacity to co-occur with protection from stressors conferred by cluster formation [6,44,45,46]. This intermediate state with both epithelial and metastatic characteristics is a known trait of CTC clusters and may promote successful survival in the bloodstream and eventual dissemination [27,34,48,49,50].

Over a 32-month period, we collected and enumerated Trop2+ (T2CTCs) and cCTCs from the blood of breast cancer patients of all subtypes, covering diverse treatment regimens, stages, and lines of metastatic therapy. 189 blood samples from 51 patients were included in the cCTC dataset, with 83 blood samples from 26 patients in the T2CTC dataset, totaling over 5000 output images to be scored. We hypothesized that Trop2, EpCAM, and CK all significantly predict CTC cluster presence and size, as well as sites of distant metastatic disease. We undertook correlative analysis and longitudinal analysis via linear mixed effects modeling (LMM) to illuminate the roles of CK8/18/19, the EpCAM family, and CD45+ cells in CTC clustering. While CELLSEARCH^®^ remains the only FDA-approved technology for breast cancer CTC enumeration, its dependence on EpCAM-based sorting means that it invariably misses EpCAM-negative CTCs [51,52]. Our aim in this study was to undertake a novel systematic analysis using the more flexible and customizable RareCyte^®^ platform in order to interrogate the effects of EpCAM, Trop2, and CK expression on CTC clustering and metastasis.

## 2. Materials and Methods

### 2.1. Study Design and Participants

Patients with metastatic breast cancer provided informed consent in accordance with IRB protocols, and all samples were anonymized prior to receipt by our technicians. Patients were heterogeneous with regard to line of therapy and time since metastatic diagnosis. Blood was collected into sodium-EDTA tubes and processed within 4 h of retrieval. Clinical characteristics for the sample populations included in all analyses of the classical and Trop2 CTC datasets are shown in Table 1 and Table 2. Included *p*-values represent Fisher’s exact tests performed with the variables in the Category column.

### 2.2. RareCyte^®^ Sample Processing, Scanning, and Analysis

Upon receipt of patient blood in EDTA tubes, blood was incubated, fractionated, and mounted per the manufacturer’s instructions. In brief, a maximum of 7.5 mL of blood was transferred to an AccuCyte^®^ Blood Collection Tube (Rarecyte, Seattle, WA, USA, #24-1160-000) for 24–48 h prior to processing. Blood was then removed from the collection tubes and dispensed into the AccuCyte Separation Tube, centrifuged to remove red blood cells, and enriched for nucleated cells. An additional centrifugation isolated nucleated cells into isolation fluid (RareCyte, Seattle, WA, USA, 24-1090-002), which were then spread as monolayers with the Cytespreader Slide Preparation Device^®^ (Rarecyte). cCTCs were stained using the RareCyte Rareplex 0700-MA protocol (Rarecyte, #24-1221-004), while T2CTCs were stained via the Rareplex 1200-MA protocol (Rarecyte, #24-1277-004) at a 1:100 concentration of mouse anti-Trop2 antibody (ECM Biosciences, Aurora, CO, USA, #TM0051, clone M005). While all primary and secondary antibody clones used in the 0700-MA are proprietary to Rarecyte, the 1200-MA assay allows incorporation of monoclonal mouse anti-Trop2 as a custom primary antibody and Rarecyte’s proprietary anti-mouse AF488 secondary. Fluorophore-biomarker pairs in the 0700-MA assay in Figure 1a were EpCAM-AF647, panCK-AF488, and CD45-Sytox Orange. The 1200-MA assay in Figure 1b,c employed EpCAM/CK-AF647, Trop2-AF488, and Sytox Orange-CD45. Slides were scanned with the Cytefinder II^®^ imager and analyzed with CyteHub^®^ software, version 3.11.1. Images were quality-checked by trained technicians to confirm classification as individual CTCs or clusters. Figure 1a,b shows representative images from 0700-MA and 1200-MA staining on HR+ and HER2-expressing cancers, respectively. Figure 1c is an exceptionally large CTC cluster from a HER2-expressing cancer, imaged via confocal microscopy. It was necessary to optimize the performance of the cCTC assay (700-MA) prior to the T2CTC assay (1200-MA). Consequently, the T2CTC dataset includes fewer samples than the cCTC dataset due to our pre-determined analysis period and the rolling enrollment characteristic of the TCCP. Additionally, while circulating cells expressing both CD45 and epithelial markers have been documented across cancer types [53,54], they were excluded from our datasets according to Rarecyte scoring protocols. Cells co-expressing CD45 and epithelial markers were excluded from the analysis.

### 2.3. Confocal Imaging

High-resolution confocal imaging of the cluster in Figure 1c was performed on the Zeiss LSM 800 Airyscan using the 63× oil objective (Zeiss, Oberkochen, Germany, 400102-9000-000).

### 2.4. Endpoints and Assessments

Our collaboration with the University of New Mexico Comprehensive Cancer Center’s Total Cancer Care protocol (TCCP) adheres to the mission statement set forward by the Oncology Research Information Exchange Network (ORIEN) [55]. The mission of the Total Cancer Care protocol is to uncover hidden trends common among cancer patients through facilitating ongoing longitudinal biological assessment of diverse patient populations, and this mission, therefore, by nature includes rolling enrollment periods and highly variable frequency of sample availability from each patient. As a consequence, a traditional clinical endpoint was not set, and instead, we applied a multi-pronged analytical approach to a finite analysis period of 32 months spanning from July 2020 through March 2023. Repeated biomarker values for each patient were averaged in the correlation analyses (Figure 2 and Figure 3) and ANOVA (Figure 4) to prevent pseudo-replication and determine biomarker co-expression patterns and effects by receptor status, respectively. In light of rolling enrollment periods and the relative rarity of TNBC, our longitudinal analyses (Figure 5, Figure 6, Figure 7 and Figure 8) were conducted relative to each patient’s initial diagnosis date with mBC (set as time 0) between HR+ and HER2+ disease only. Longitudinal analysis by metastatic site was conducted relative to the first diagnosis dates for each unique metastatic site. Because of the diversity of treatment regimens and rolling patient enrollment characteristic of the Total Cancer Care protocol, we aimed to interrogate the underlying patterns of CTC shedding after metastatic diagnosis as functions of receptor subtype and distinct metastatic sites that persist independent of treatment history. Successful disease management by definition improves prognosis [56,57], while increased CTC counts are predictive of poor prognosis [21,25,45]. Paradoxically, however, all widely applied treatment regimens can themselves promote CTC shedding [58,59,60]. Accordingly, while the treatment regimens applied to patients in our datasets may affect CTC presence and clustering, each regimen has the potential to increase or decrease CTC counts in ways peculiar to each individual’s unique history [58,59,60,61]. Our multi-pronged statistical approaches therefore detect patterns of CTC biomarker expression that persist irrespective of treatment.

### 2.5. Statistical Analyses

Due to non-normality and zero-skewing in the patient biomarker datasets, data were base 2 log-transformed after the addition of 1 to each original observation. Correlation coefficient (r) values are listed, and correlation analysis was performed alongside linear regressions for all biomarker pairs, as shown in Figure 2 and Figure 3. The classical CTC (cCTC) and Trop2+ CTC (T2CTC) dataset each evaluated 8 biomarkers; however, the cCTC dataset includes EpCAM and pan-CK on different fluorophores, but the T2CTC dataset does not because the 1200-MA staining requires that EpCAM and CK be measured with the same fluorophore. Longitudinal data analyses were performed based on a linear regression model with random coefficients according to the formula **Y = a + b x t + ε**, where **a** represents the initial biomarker value, decomposed into a fixed effect **a_0_**, and a random effect **a_1_**, while **b** represents the slope, decomposed into **b_0_** and **b_1_** as above. Both datasets were fitted to the LMM described above to determine if biomarker trajectories differed by receptor status (HR+ and HER2+), as there were not sufficient patients or collection events to include TNBC in the longitudinal analyses. For longitudinal analysis, the complete cCTC and T2CTC datasets were fit to two separate linear models. In the full model, the intercept **a_0_** and slope **b_0_** differ between the two receptor statuses. The full model tests whether HR+ and HER2+ disease differ in their CTC clustering biomarker trajectories after the date of first diagnosed metastasis. Where evidence was insufficient to detect differences by receptor status, data was fit to a reduced model to determine if the slope was significantly non-zero for each biomarker value over the analysis period. In the reduced model, HR+ and HER2+ diseases are analyzed together, and the model tests whether the biomarker slope is significantly non-zero after metastatic diagnosis, irrespective of disease subtype. This is to determine whether metastatic diagnosis per se is associated with an increase in biomarkers related to CTC clustering. In the reduced model, the intercept **a_0_** but not the slope **b_0_** differ by receptor status. Longitudinal analyses were performed with the date of each patient’s first diagnosis with metastatic breast cancer standardized as time 0. The patient characteristics for the 51 individuals included in correlation analysis (Figure 2 and Figure 3), ANOVA (Figure 4), and the full and reduced LMMs (Figure 5, Figure 6, Figure 7 and Figure 8) are shown in Table 3 and Table 4. The complete results and parameters of the LMM (i.e., the linear regression model with random coefficients), including the estimates for the intercepts, slopes of the trend lines, and the differences between the trend lines, can be found in the Appendix A. For longitudinal analysis by metastatic site, time 0 represents each patient’s diagnosis with metastasis to the indicated site. All statistical analysis was performed in R version 4.5.0 using the lmerTest and lme4 packages. Data visualization for Figure 4 was performed in Graphpad Prism version 10.4.1. We used the markdown-file tool implemented in R for coding and creating dynamic documentation in statistical analysis to ensure scientific rigor and reproducibility.

## 3. Results

### 3.1. RareCyte Reveals Expression of Trop2 in Breast Cancer Patient CTCs and High Inter-Marker Correlation

Analysis of breast cancer patient blood by RareCyte technology revealed the clear presence of cCTCs (EpCAM+, PanCK+, CD45- cells, Figure 1a) as well as the presence of T2CTCs (Figure 1b,c). Additionally, both classical and T2CTCs were found as singlets and in homotypic and heterotypic cluster configurations (Figure 1a–c).

As expected, correlation analysis revealed a high degree of correlation among a majority of the included CTC biomarkers. Figure 2 and Figure 3 list biomarkers in the diagonals, with each box on the lower left representing scatter plots with the regression line of the two biomarkers in the same row and column, and each box in the upper right representing the correlation coefficient of the same two biomarkers in that same row and column, symmetrically about the diagonal. The tables included in both figures list all strong correlations (those with an r ≥ 0.7). In the cCTC dataset (Figure 2), expression of EpCAM and the presence of CD45+ cells in clusters with CTCs were highly predictive of both cluster presence and size (highlighted in red text), with a correlation coefficient of 0.78 between EpCAM+ CTCs and the presence of CD45+ cells. No strong correlations were observed between pan-Cytokeratin expression and any of the other evaluated biomarkers. Likewise, in the T2CTC dataset, cluster presence and size were highly correlated with the presence of CD45+ cells, as well as positivity for the CK/EpCAM channel jointly with Trop2 (Figure 3). Together, this data strongly suggests that the EpCAM family and CD45+ cells are facilitative of CTC cluster formation and intravasation to the blood of breast cancer patients.

### 3.2. No Individual CTC Biomarker Condition Was Significantly Associated with Receptor Status

To determine whether patterns of individual biomarker expression were associated with receptor status, we compared biomarker values across all patients in both the cCTC (Figure 4a–d) and T2CTC datasets (Figure 4e,f). In cases where patients had more than 1 sample analyzed, each patient is represented by the average value for each biomarker to avoid pseudo-replication. ANOVA revealed all associations to be non-significant. Results are shown in Table 1 (cCTC) and Table 2 (T2CTC).

### 3.3. Longitudinal Analysis Reveals Differences in CTC Clustering Between HR+ and HER2+ Cancers

To determine whether biomarkers changed over the course of multiple measures, we performed longitudinal analysis on HR+ and HER2+ patients for both the cCTC (Figure 5 and T2CTC datasets (Figure 6). Figure 5a–c show spaghetti plots [62] with trendlines for clustering in the cCTC dataset, with those for EpCAM in Figure 5d. Full LMM results for longitudinal analysis by receptor status are shown for the cCTC dataset in Table 3, with those of the T2CTC dataset in Table 4.

EpCAM and the number of clusters containing 2 or more cells (Clusters > 2) showed statistically significant differences (*p* = 0.022 and *p* = 0.007, respectively) in slope between HR+ and HER2+ cancers, indicating different biomarker rates of change over time between the two receptor subtypes. In particular, analysis indicates that these two parameters increased over time in HER2+ cancers during the analysis period relative to the first metastatic diagnosis, while there is no significant change with time since metastasis in HR+ cancers. Clusters showed marginal significance (*p* = 0.068) in the same direction, and all other comparisons were non-significant.

Figure 6a–c show spaghetti plots with trendlines for clustering in the T2CTC dataset, with those for Trop2 in Figure 6d. Full LMM results are shown in Table 4. All comparisons by receptor status were non-significant. To determine if CTC biomarker trajectories in HR+ and HER2+ disease differed by metastatic site, independent LMM were employed for each of four metastatic sites: brain, liver, bone, and lungs. The model for each metastatic site included analysis of all biomarkers, with clustering and EpCAM-family expression shown in Figure 7 for the cCTC dataset and in Figure 8 for the T2CTC dataset. Complete results for the full and reduced models for each metastatic site in the cCTC dataset are shown in Appendix A and T2CTCs in Appendix A. In the cCTC dataset, the full model found no significant differences between HR+ and HER2+ cancers by metastatic site. The cCTC reduced model found that relative to the first diagnosis of lung metastasis, Clusters > 2 (*p* = 0.072), CD45 in cluster (*p* = 0.07763) had marginally non-zero slopes. Relative to diagnosis with bone metastasis, slopes for Clusters > 2 (*p* = 0.070) and CD45 in cluster (0.091) were marginally non-zero. Relative to diagnosis with brain metastasis, Clusters > 2 (0.080) were marginally non-zero. Similar results were obtained for liver metastasis: Clusters > 2 (*p* = 0.086).

In the T2CTC dataset (Figure 8), a baseline effect of Trop2 expression was found for the bone metastatic condition (*p* = 0.04255), with a marginally significant difference in slope between those with bone metastasis and without (*p* = 0.052). Clusters > 2 were significant at baseline (*p* = 0.009), and with respect to slope (*p* = 0.021), indicating an increase in larger clusters over time after diagnosis with brain metastasis. The reduced model found marginally non-zero slopes by lung metastasis for Clusters (*p* = 0.085) and by brain metastasis for CD45 in clusters (*p* = 0.076). All other comparisons were non-significant.

## 4. Discussion

Due to their roles as epithelial cell adhesion molecules, the two members of the EpCAM family have long been suspected to play fundamental roles in CTC invasion, intravasation, circulation, dissemination, and metastasis. Notably, their mechanisms of action during tumor progression have remained elusive [31,32,35,40,63]. EpCAM is employed as the primary discriminating diagnostic marker in breast cancers, and decades of study have confirmed its salience in predicting overall survival and progression-free survival heterogeneity. Due to the widely acknowledged need to expand biomarker criteria in CTC identification, we incorporated EpCAM homolog Trop2 in our analyses. Consistent with their distinct known molecular functions [37,38,39,40,41], our results indicate that Trop2 and EpCAM have overlapping but distinct roles in CTC shedding, clustering (Figure 5 and Figure 6), and homing to distant metastatic sites (Figure 7 and Figure 8). In support of this hypothesis, both EpCAM+ CTCs and Clusters > 2 increased after the first metastasis in HER2+ but not HR+ cancers (Figure 5b,d and Appendix A), suggesting discrete evolutionary trajectories in CTC clustering between the receptor statuses. Additionally, correlation analysis of cCTCs revealed that EpCAM strongly correlated with cluster presence, cluster size, and the presence of CD45+ cells in the cluster, but not the number of CTCs in the cluster (Figure 2). Analyses of T2CTCs revealed that expression of CK/EpCAM alongside Trop2, but not CK/EpCAM alone, correlated strongly with the number of CTCs in the cluster, or greater homotypic adhesion (Figure 3). These results suggest that while both EpCAM and Trop2 contribute to cluster formation, EpCAM expression may facilitate a bias toward heterotypic clustering with CD45+ cells, while Trop2 may promote homotypic clustering with other CTCs. These strong correlations are highly suggestive of discrete contributions of Trop2 and EpCAM to CTC clustering.

While the full cCTC model comparing trajectories of biomarkers between HR+ and HER2+ disease did not find any significant association between metastatic site and biomarker status, the full T2CTC model did find a significant baseline effect of Trop2 expression, with a marginally significant difference in slope between receptor statuses after diagnosis with bone metastasis. While the smaller size of the T2CTC dataset relative to the cCTC dataset does warrant caution, the results suggest continued shedding of Trop2+ CTCs may be a feature of bone metastatic disease (Figure 8d and Appendix A); however, we did not find baseline evidence of distinct Trop2 enrichment between HR+ and HER2+ disease (Figure 4d). Additionally, in the T2CTC dataset, larger clusters (those with greater than 2 cells) differed both at baseline and by slope between receptor subtypes after diagnosis with brain metastasis (Figure 8b and Appendix A), suggesting that larger cluster size corresponds more to brain metastasis in HER2+ than in HR+ disease. This has critical potential clinical relevance, as the prevention of overt brain metastasis development in HER2+ breast cancers is a universally recognized unmet need. The use of CTC trends to predict risk of brain metastasis development may be clinically actionable with development of multiple new HER2-targeted agents with central nervous system (CNS) activity, including tucatinib and trastuzumab deruxtecan [64]. Intervening early against the rise in large clusters with one of these agents may eradicate microscopic metastatic disease to the brain prior to the development of overt metastasis, thus delaying time to clinical CNS progression.

The reduced cCTC models by metastatic status revealed marginally non-zero slopes for Clusters > 2 and clustering with CD45+ cells after diagnosis with lung bone metastasis, as well as for Clusters > 2 for brain and liver metastasis. The reduced T2CTC model found marginally non-zero slopes for clusters present after diagnosis with lung metastasis and CD45+ immune cells in clusters after brain metastasis.

Taken together, modeling by metastatic site suggests that certain sites are predisposed to shedding larger heterotypic clusters containing CD45+ cells. Larger clusters are known to be associated with poor prognosis [34,65], and prior research from our group has demonstrated a transcriptomic signature underlying breast cancer brain metastasis [52]. Importantly, to our knowledge, this is the first study using patient-derived CTCs to tie specific CTC cluster configurations with specific organ metastatic sites [66]. Although recent investigations of organotropism by breast cancer subtypes have been performed with cell lines using microfluidic chips. For example, SKBR3, a Trop2-high, HER2-expressing cell line, did not show substantial bone organotropism [67].

Contrary to our hypothesis, while the pan-cytokeratin marker targeting CK8/9/19 was detected in many patients, it did not correlate strongly (r > 0.7) with any other CTC biomarker or clustering. This is surprising given the role of CKs in maintenance of cell morphology and, in particular, the role of CK19 in cell–cell adhesion. However, this mechanism is mediated by E-Cadherin, which is frequently lost in the partial EMT state [20,21]. Because interactions between CK19 and E-cadherin result in stronger cell adhesion and cell polarization that may impede clustering and migration, the relationships between clustering and EpCAM and Trop2 may be due to their ability to mediate both homophilic adhesion and tight-junction mediated adhesion. This allows simultaneously for cytoskeletal flexibility, maintenance of a depolarized state, and preservation of cell–cell contacts. EpCAM+ CTCs expressing low levels of CK/8/18/19 correspond with decreased overall survival [21,68]. Our results suggest that this could be a function of clustering in the hybrid EMT state. The hypothesized dual role in clustering is consistent with the partial/hybrid EMT phenotype, which may confer the plasticity found in the most metastatically competent CTC clusters [5,27,28,34,35,65,69], and further underscores the complexity of oncogenic patterns of differentiation and dedifferentiation. While CK19-KO cells have improved motility, they are less able to form tumor mammospheres [22], suggesting a complex role whereby they are not integral to invasion and cluster formation, but may aid in guiding structured micrometastasis upon dissemination.

Our analysis of biomarker values by subtype did not reveal any significant associations, although all biomarker values for a given patient were averaged in this analysis to avoid pseudoreplication. Given the high inter- and intra-patient heterogeneity in circulating EpCAM+ cells shown here (Figure 5 and Figure 6) and elsewhere [42], use of averaged values may have obscured underlying patterns that differed by receptor subtype.

Conversely, our longitudinal analysis, which incorporates intra-patient heterogeneity in CTC biomarkers, revealed that biomarker trajectories did in fact differ significantly between subtypes (Figure 5 and Table 3). Furthermore, as previous investigations yielded conflicting reports as to how CTC number and presence differ between subtypes [25,28,29,30,42], we suggest that this disparity may be in part explained by high intra-patient heterogeneity, discrete biomarker trajectories between subtypes, and unaccounted for EpCAM-negative CTCs [26].

Correlational analysis also revealed that CD45+ cell presence in clusters correlated with cluster size, independent of EpCAM and Trop2 expression. While CD45+ cells may be more likely to develop connections to larger clusters in a probabilistic manner, and the EpCAM family may have an indirect effect on CD45+ cells clustering with tumor cells mediated by greater cluster size, our analysis cannot establish a causal link between CD45+ cells in clusters with EpCAM+ and Trop2+ CTCs. However, EpCAM is capable of mediating homotypic interactions between the intestinal epithelia and epithelia-resident lymphocytes [18,70], which is highly suggestive of a possible role in cluster formation prior to dissemination.

There are limitations to this study. First, the smaller size of the T2CTC dataset and the marginal significance of the results by metastatic site merit caution. However, given the consistency of the effect patterns despite the substantial confounds due to tumor heterogeneity and treatment status, we contend these effects reveal an underlying mechanism. Second, while our data suggest that EpCAM and Trop2 play roles in the presence and size of CTC clusters, there are other known biomarkers involved in cluster formation unaccounted for in our dataset, including plakoglobin [6], CD44 [69], and ICAM [69]. CK19′s interactions with plakoglobin and E-Cadherin are complex and context-dependent, further complicating interpretation. Third, some known bidirectional molecular interactions such as those between HER2 and CK19 [21,71] are not analyzable in our datasets due to our use of the pan-CK marker, which stains CK8, CK18, and CK19. Lastly, CK19 may serve to impair or facilitate metastasis depending on cancer stage [22].

Altogether, our results illustrate the critical roles of Trop2 and EpCAM in CTC clustering (Figure 2 and Figure 3), and reveal discrete CTC biomarker trajectories between HR+ and HER2+ disease (Figure 5) after metastatic diagnosis, giving clarity to a clinical literature largely confounded by inter- and intra-patient heterogeneity in CTC shedding and clustering. Further studies are needed to interrogate the roles of Trop2 and EpCAM. In particular, whether their actions in Claudin stabilization and homophilic adhesion differentially regulate CTC cluster formation and metastatic competency.

## 5. Conclusions

CTC shedding is a complex process. Given the rarity of CTCs and the further rarity of metastatically competent CTCs and CTC clusters, our systematic analyses represent an important step in filling clinically actionable gaps of knowledge. Correlation and longitudinal analyses revealed that EpCAM, Trop2, and CD45 expression were highly predictive of cluster presence and size (Figure 2 and Figure 3). We conclude from our correlation analyses that Trop2 and EpCAM facilitate cluster formation, while CK 8/18/19 are less critical to CTC clustering. We also highlighted distinct trajectories of biomarker change after metastatic diagnosis between HR+ and HER2+ cancers (Figure 5). Additionally, our analyses incorporating Trop2 show that cluster size increases after brain metastatic diagnosis in HER2+ but not HR+ disease (Figure 8b). We conclude from these results that HER2+ cancers stand to benefit considerably from early application of therapeutics with CNS activity such as tucatinib and trastuzumab deruxtecan. By incorporating analysis of EpCAM, Trop2, and CK8/18/19, we were able to shed light on the roles of these CTC markers in clustering and metastatic trajectories, finding that EpCAM and Trop2, but not CK, significantly affected clustering and distant metastasis. Future studies building on our research may illuminate the mechanistic roles of EpCAM, Trop2, and the partial EMT in CTC clustering and organotropism, in turn yielding clinically actionable early treatment strategies to undermine the competency of these deadly metastatic seeds.

## Figures and Tables

**Figure 1 cancers-17-02717-f001:**
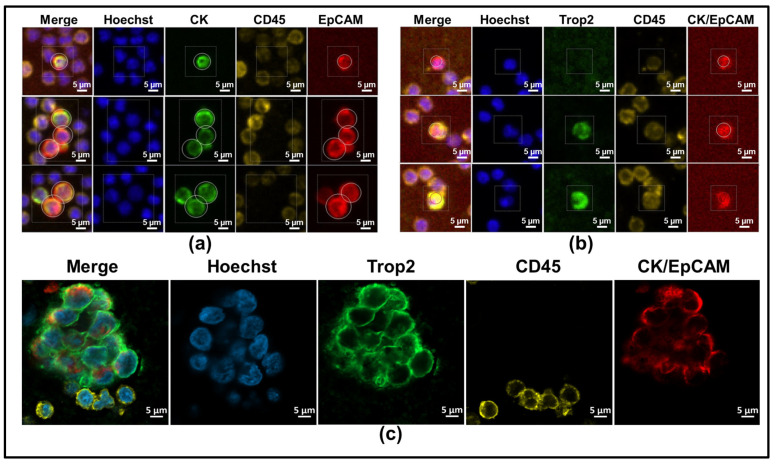
Rarecyte reveals expression of Trop2 in breast cancer patient CTCs. (**a**,**b**) Immunofluorescence identification of CTCs using the Cytefinder II platform. (**a**) cCTCs identified by pan-Cytokeratin (green, AF488) and EpCAM (red, AF647) positive staining were visualized as individual cells or in small clusters with CD45+ cells (yellow, Sytox Orange). (**b**) Trop2+ CTCs were identified by Trop2 (green, AF488) and a joint CK/EpCAM channel (red, AF647) (**a**). (**c**) Confocal image of an exceptionally large cluster showing substantial CTC-CTC clustering and interfacing with CD45+ immune cells.

**Figure 2 cancers-17-02717-f002:**
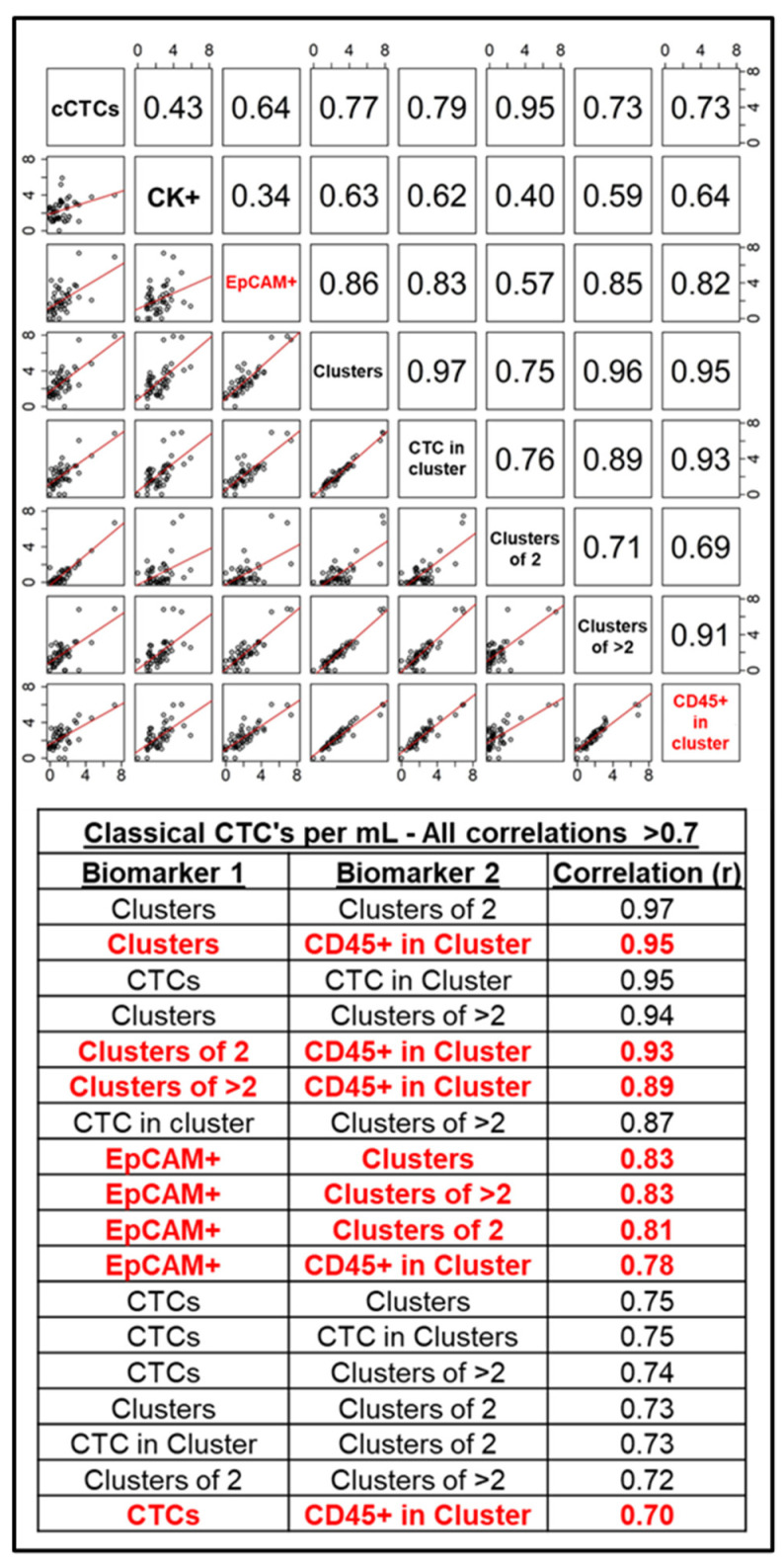
Correlation analysis of classical CTC biomarkers. **Top panel**: Biomarkers are listed diagonally from the top left box down to the bottom right. Scatter plots with linear regression lines for each biomarker pair are in the bottom left, with correlation coefficients for each biomarker pair in the top right, symmetrically about the diagonal. **Bottom panel**: All biomarker pairs that yielded a correlation coefficient of greater than 0.7, indicating a strong correlation. EpCAM and CD45 in Cluster are in red for ease of reference.

**Figure 3 cancers-17-02717-f003:**
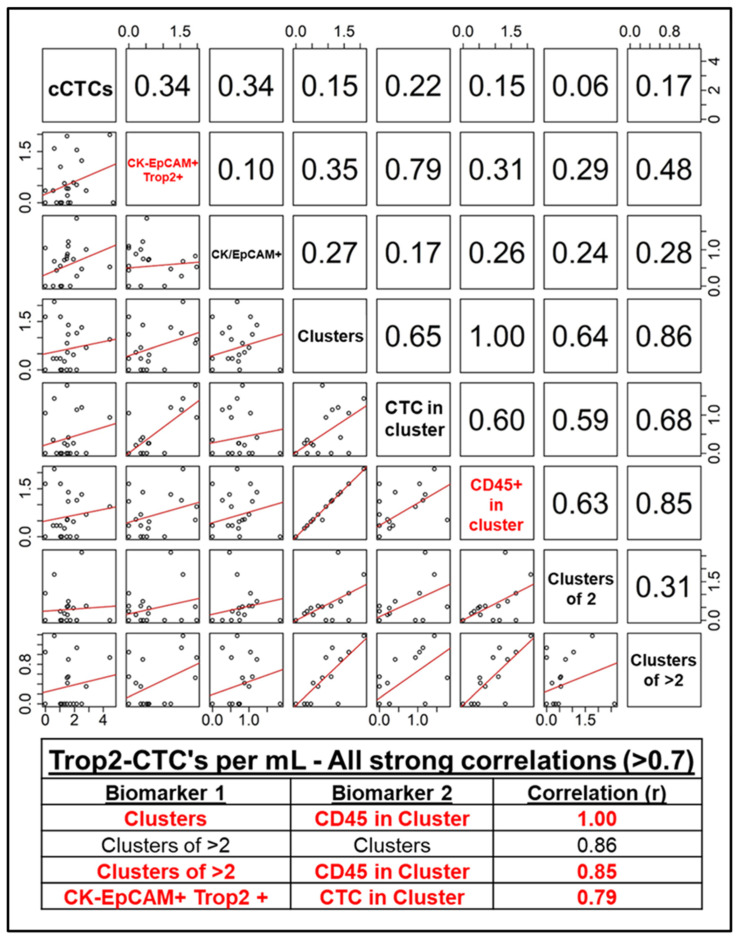
Correlation analysis of Trop2-CTC Biomarkers. **Top panel**: Biomarkers are listed diagonally from the top left box down to the bottom right. Scatter plots with linear regression lines for each biomarker pair are in the bottom left, with correlation coefficients for each biomarker pair in the top right, symmetrically about the diagonal. **Bottom panel**: All biomarker pairs that yielded a correlation coefficient of greater than 0.7, indicating a strong correlation. CK-EpCAM+/Trop2+ and CD45 in Cluster are in red for ease of reference.

**Figure 4 cancers-17-02717-f004:**
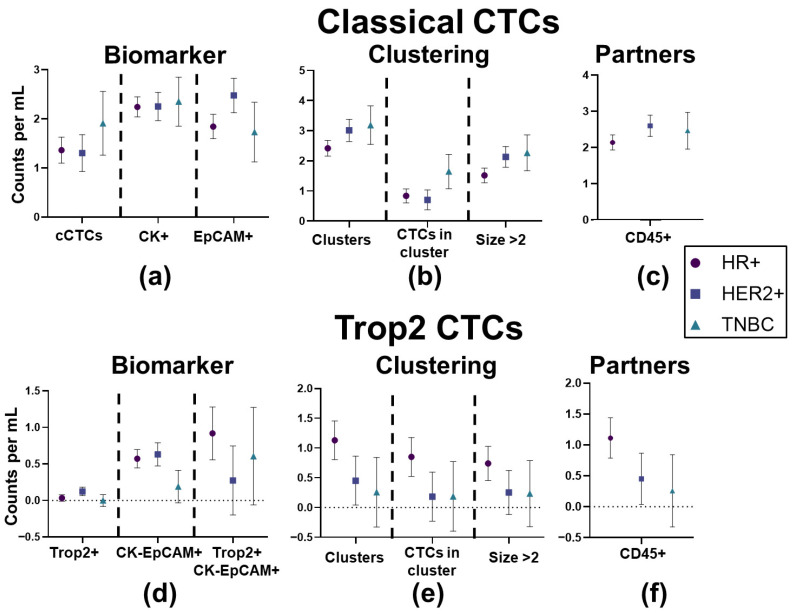
(**a**–**c**) Biomarkers of CTC presence and clustering as measured in the classical CTC dataset, while (**d**–**f**) show those from the Trop2 dataset. All between-receptor status differences were non-significant by ANOVA.

**Figure 5 cancers-17-02717-f005:**
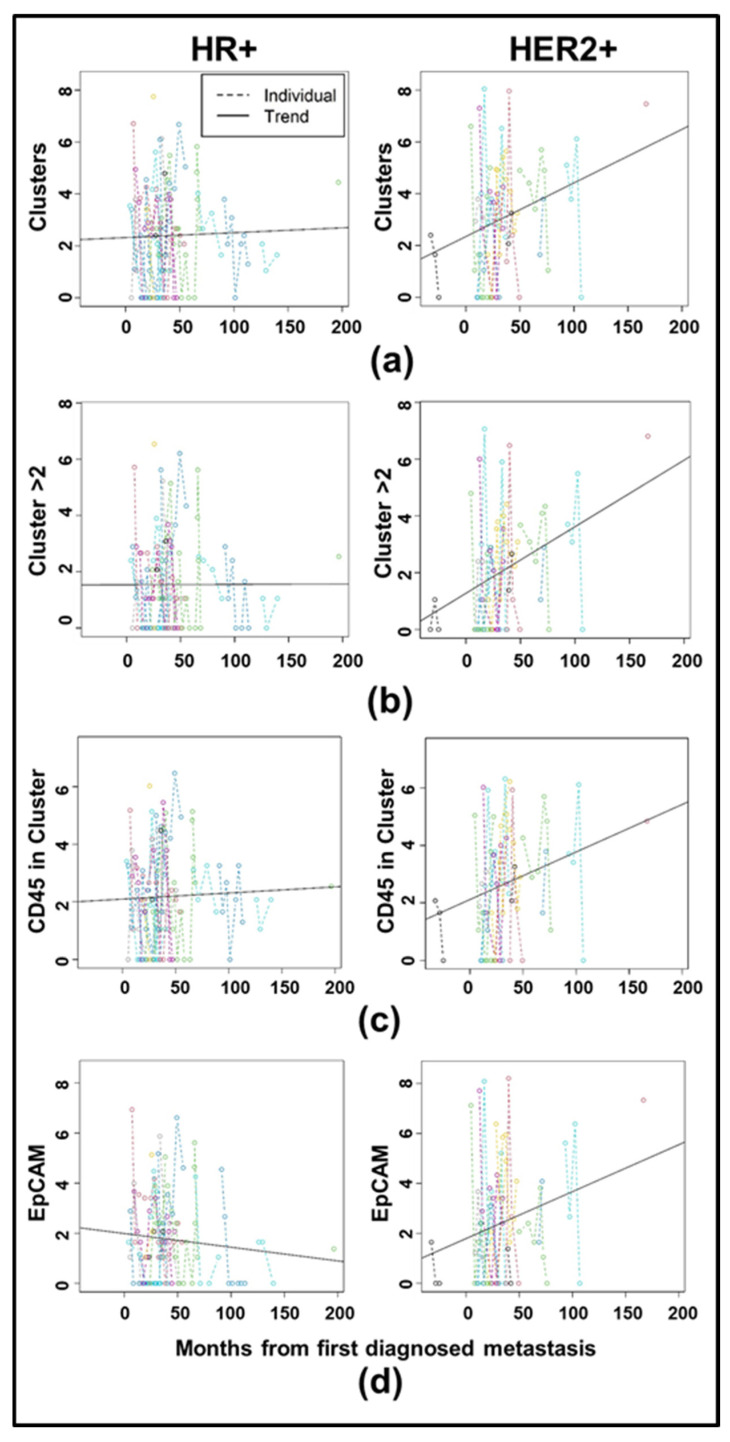
Spaghetti plots with trendlines for cCTC clustering in HR+ versus HER2+ cancers, with the first diagnosed metastasis as time 0: (**a**) Number of clusters detected for HR+ and HER2+ cancers, respectively (*p* = 0.06746). (**b**) Number of clusters detected with more than two cells. (**c**) Number of clusters containing CD45+ cells. (**d**) Number of EpCAM+ cells detected. The slope over the analysis period for Clusters > 2 and EpCAM were significantly different (*p* = 0.00691 and *p* = 0.02184), indicating that these two parameters increased in HER2+ cancers after diagnosis, but did not increase in HR+ disease. All other results were non-significant, indicating no difference in slope between receptor statuses. All data are per mL of blood.

**Figure 6 cancers-17-02717-f006:**
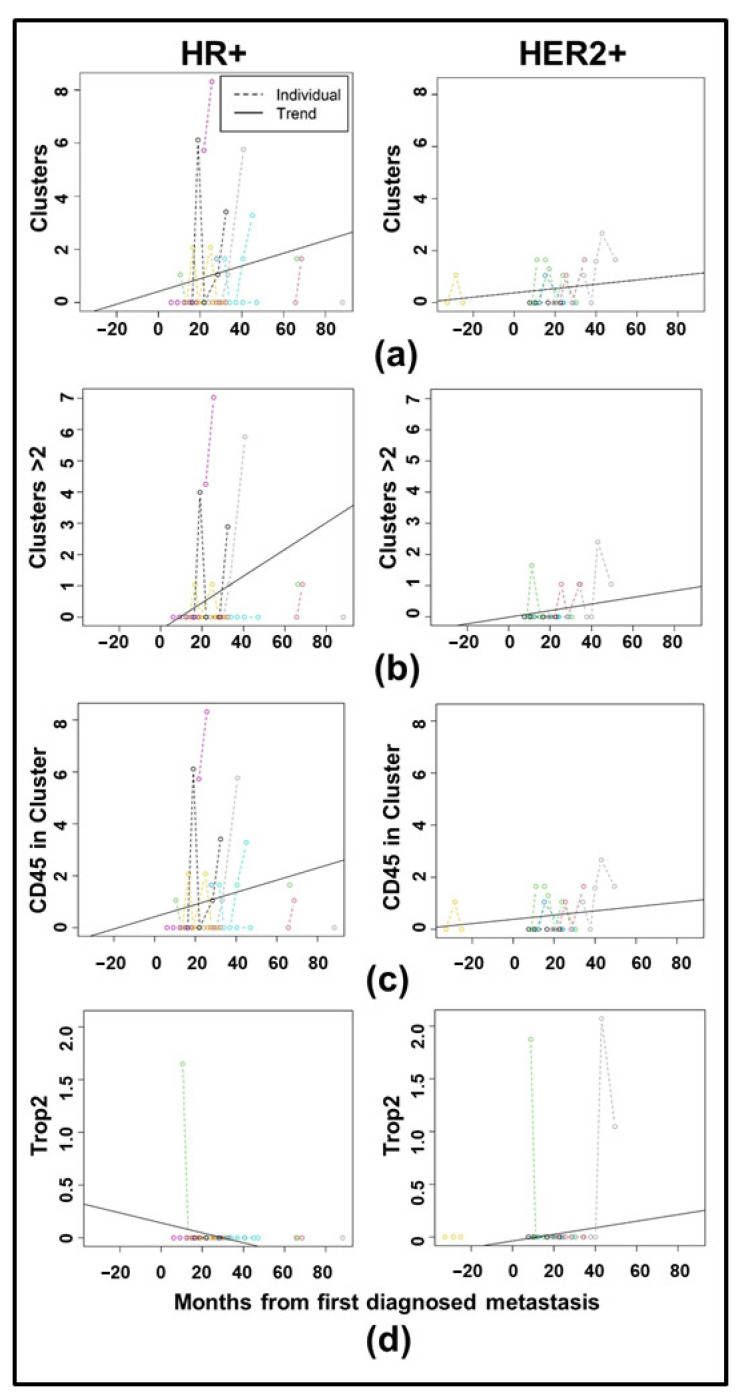
Spaghetti plots with trendlines for T2CTC clustering over the course of the analysis period for HR+ versus HER2+ cancers, centered on the first diagnosis of metastasis as time 0: (**a**) Number of clusters detected for HR+ and HER2+ cancers, respectively. (**b**) Number of clusters detected with greater than two cells. (**c**) Number of clusters containing CD45+ cells. (**d**) Number of Trop2+ cells detected. All comparisons were non-significant, indicating no difference in slope between receptor statuses. All data are per mL of blood.

**Figure 7 cancers-17-02717-f007:**
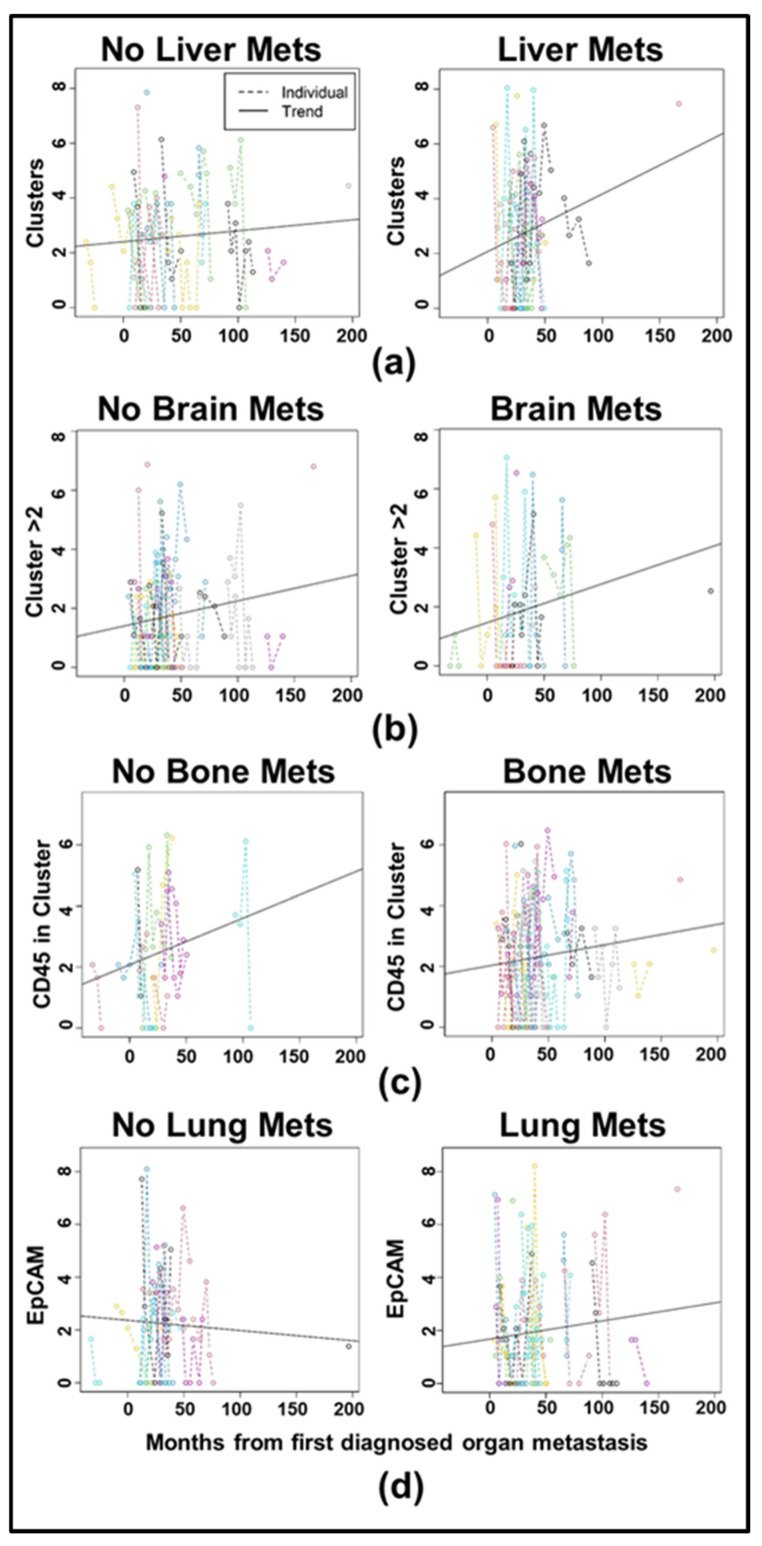
Spaghetti plots with trendlines for cCTC clustering over the course of the analysis period by select metastatic site: (**a**) Number of clusters detected in patients without versus with liver metastasis, in the left and right panels, respectively. (**b**) Number of clusters detected with greater than two cells in patients without versus with brain metastasis. (**c**) Number of clusters containing CD45+ cells in patients without versus with bone metastasis. (**d**) Number of EpCAM+ cells detected in those without versus with lung metastasis. All data are per mL of blood.

**Figure 8 cancers-17-02717-f008:**
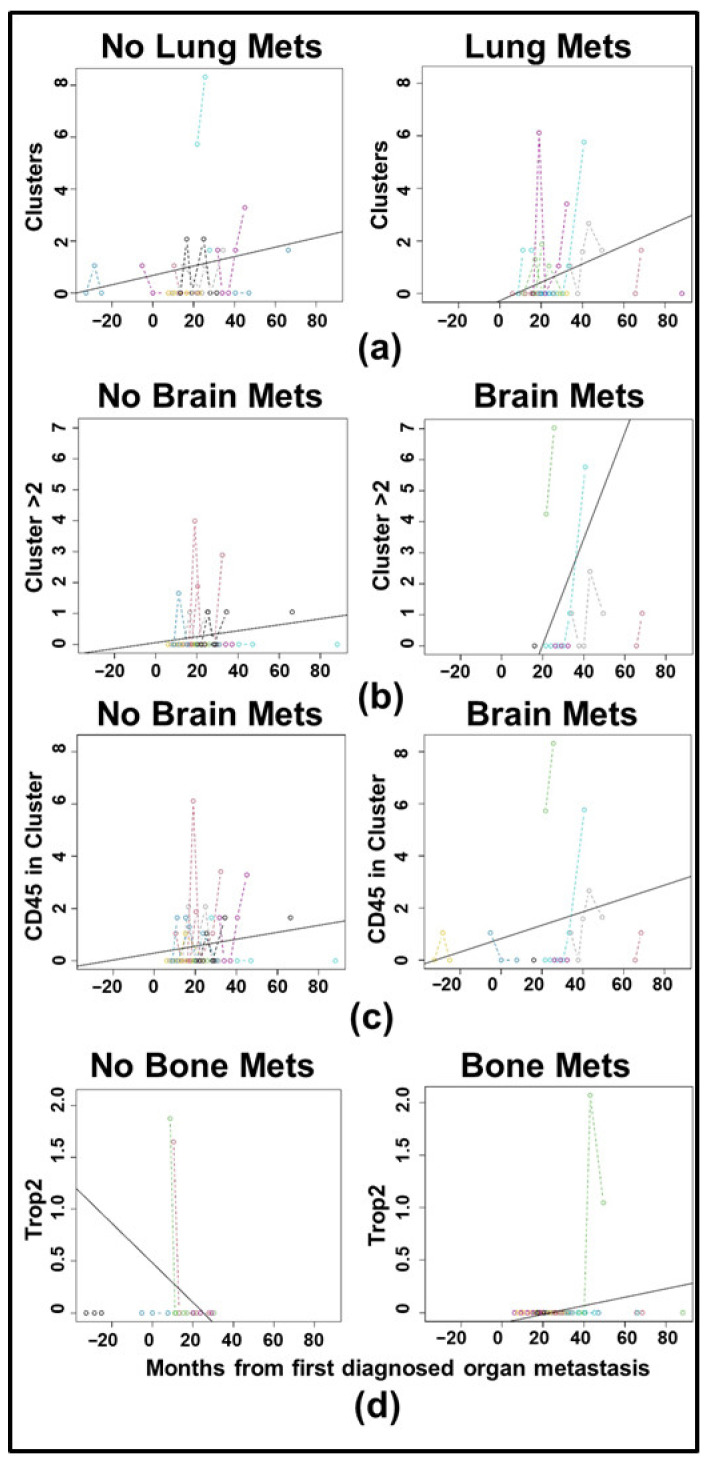
Spaghetti plots with trendlines for T2CTC clustering over the course of the analysis period by select metastatic site: (**a**) Number of clusters detected in patients without versus with lung metastasis, in the left and right panels, respectively. (**b**) Number of clusters detected with greater than two cells in patients without versus with brain metastasis. (**c**) Number of clusters containing CD45+ cells in patients without versus with brain metastasis. (**d**) Number of Trop2+ cells detected in those without versus with bone metastasis. All data are per mL of blood.

**Table 1 cancers-17-02717-t001:** Clinical characteristics of patients included in the classical CTC dataset.

Clinical Characteristic	Category	Full Cohort	HR+/HER2-	HER2+	TNBC	Fisher Exact Test, *p*-Value
Age at 1st blood collection	<65	30 (60.0)	17 (54.8)	11 (68.8)	2 (66.7)	
	65+	20 (40.0)	14 (45.2)	5 (31.2)	1 (33.3)	0.792
	Total	50 (100)	31 (100)	16 (100)	3 (100)	
Number of Metastatic sites	1	4 (8.0)	4 (12.9)	0 (0.0)	0 (0.0)	0.417
	2	15 (30.0)	7 (22.6)	7 (43.8)	1 (33.0)	
	3+	31 (62.0)	20 (64.5)	9 (56.2)	2 (66.7)	
	Total	50 (100)	31 (100)	16 (100)	3 (100)	
Lung metastasis	N	22 (43.1)	14 (43.8)	7 (43.8)	1 (33.0)	1.000
	Y	29 (56.9)	18 (56.2)	9 (56.2)	2 (66.7)	
	Total	51 (100)	32 (100)	16 (100)	3 (100)	
Bone metastasis	N	14 (27.5)	6 (18.8)	7 (43.8)	1 (33.0)	0.141
	Y	37 (72.5)	26 (81.2)	9 (56.2)	2 (66.7)	
	Total	51 (100)	32 (100)	16 (100)	3 (100)	
Liver metastasis	N	25 (50.0)	16 (51.6)	7 (43.8)	2 (66.7)	
	Y	25 (50.0)	15 (48.4)	9 (56.2)	1 (33.3)	0.816
	Total	50 (100)	31 (100)	16 (100)	3 (100)	
Brain metastasis	N	34 (68.0)	24 (77.4)	9 (56.2)	1 (33.3)	
	Y	16 (32.0)	7 (22.6)	7 (43.8)	2 (66.7)	0.105
	Total	50 (100)	31 (100)	16 (100)	3 (100)	

Of the 51 patients in the cCTC dataset, 1 was missing age, and 1 did not have full metastatic site data.

**Table 2 cancers-17-02717-t002:** Clinical characteristics of patients included Trop2 CTC dataset.

Clinical Characteristic	Category	Full Cohort	HR+/HER2-	HER2+	TNBC	Fisher Exact Test, *p*-Value
Age at 1st blood collection	<65	16 (64.0)	7 (50.0)	7 (87.5)	2 (66.7)	
	65+	9 (36.0)	7 (50.0)	1 (12.5)	1 (33.3)	0.246
	Total	25 * (100)	14 (100)	8 (100)	3 (100)	
Number of Metastatic sites	1	2 (7.7)	2 (13.3)	0 (0.0)	0 (0.0)	0.834
	2	9 (34.6)	4 (26.7)	4 (50.0)	1 (33.0)	
	3+	15 (57.7)	9 (60.0)	4 (50.0)	2 (66.7)	
	Total	26 (100)	15 (100))	8 (100)	3 (100)	
Lung metastasis	N	13 (50.0)	8 (53.3)	4 (50.0)	1 (33.0)	1.000
	Y	13 (50.0)	7 (46.7)	4 (50.0)	2 (66.7)	
	Total	26 (100)	15 (100)	8 (100)	3 (100)	
Bone metastasis	N	7 (26.9)	2 (13.3)	4 (50.0)	1 (33.0)	0.139
	Y	19 (73.1)	13 (86.7)	4 (50.0)	2 (66.7)	
	Total	26 (100)	15 (100)	8 (100)	3 (100)	
Liver metastasis	N	11 (42.3)	6 (40.0)	3 (37.5)	2 (66.7)	
	Y	15 (57.7)	9 (60.0)	5 (62.5)	1 (33.3)	0.724
	Total	26 (100)	15 (100)	8 (100)	3 (100)	
Brain metastasis	N	18 (69.2)	11 (73.3)	6 (75.0)	1 (33.3)	
	Y	8 (30.8)	4 (26.7)	2 (25.0)	2 (66.7)	0.418
	Total	26 (100)	15 (100)	8 (100)	3 (100)	

* Of the 26 total patients included in the T2CTC dataset, 1 did not have a listed age.

**Table 3 cancers-17-02717-t003:** Full LMM results of longitudinal analysis by receptor status for the cCTC dataset.

Biomarker (per mL)	Receptor Effect	Time Effect	Interaction
cCTCs	0.6591061	0.7461176	0.840692
CK	0.4685809	0.6062315	0.4074604
** EpCAM **	0.7158591	0.1999038	** 0.0218357 **
**Clusters**	0.9510659	0.028952	**0.0674635 ^†^**
Clusters of 2	0.6826174	0.0805827	0.1121893
cCTC in cluster	0.4323639	0.5820074	0.5686966
**Clusters > 2**	0.5347161	0.0062573	** 0.0069147 **
CD45 in cluster	0.9598183	0.0401535	0.1091464

**^†^** Indicates a marginally significant result. Significant results are in red, marginally significant results are in bold.

**Table 4 cancers-17-02717-t004:** Full LMM model results of longitudinal analysis by receptor status for T2CTC dataset.

Biomarker (per mL)	Receptor Effect	Time Effect	Interaction
CK/EpCAM/Trop2	0.4802839	0.8340946	0.8919177
CK/EpCAM	0.5649867	0.0955775	0.1789835
Trop2	0.5484958	0.8870074	0.490742
Clusters	0.966292	0.2493243	0.5708911
CTC in cluster	0.7295486	0.5810308	0.8777693
CD45 in cluster	0.9998403	0.2495356	0.5364823
Clusters of 2	0.2829963	0.6461081	0.4631998
Clusters > 2	0.633026	0.2716701	0.4975725

## Data Availability

Data are available upon request but are not publicly available for protection of patient privacy.

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
