# Peer review of "Comprehensive Longitudinal Linear Mixed Modeling of CTCs Illuminates the Role of Trop2, EpCAM, and CD45 in CTC Clustering and Metastasis"

_cancers, 2025, doi:10.3390/cancers17162717_

Round 1
Reviewer 1 Report
Comments and Suggestions for Authors
The manuscript "Comprehensive longitudinal linear mixed modelling of CTCs illuminates the role of Trop2, EpCAM, and CD45 in CTC clustering and metastasis", written by Merkley et al., aims on the role of selected factors in clustering of circulating tumor cells and in metastasises. The research project is well defined, experimental setup is well selected. The Introduction is clearly written, however, the novelty of the presented approach should be more highlighted in the last two paragraphs, where discussing HER2 and Trop2. The experimental section is well covered, and all methods adequately described. The results part is also well described and the text carefully prepared. The conclusion part should be more specific on what the results indicate and what is the added value this research brings into the scientific topic of breast cancer diagnosis/treatment.
I have few comments regarding the figures and statistical analysis:
- Figures are not well plotted, and the graphics fall behind current standards
- Text is very small and unreadable in many cases
- Use of the foint-size to show significance of the value is in general great idea; however, some fonts are not readable in figures 2 and 3.
- Figures 5 - 8 are almost impossible to read and thus they are not understandable
- Authors should optimize ranges in all X-Axis to fit the data; there is no need from large blank spaces in the plots
- The authors should show also parameters of the particular regression models for the trends plotted in the figures 5 - 8.
Author Response
Reviewer #1
…the novelty of the presented approach should be more highlighted in the last two paragraphs [of the Introduction], where discussing HER2 and Trop2.
RESPONSE: We thank the reviewer for their comments; and have revised Introduction accordingly. Please see lines 148-153.
…conclusion should be more specific on what the results indicate and what is the added value this research brings into of breast cancer diagnosis/treatment.
RESPONSE: We have revised and expanded the conclusions in line with reviewer suggestions. Please see lines 502-511.
- Figures are not well plotted, and the graphics fall behind current standards
RESPONSE: We have revised Figures 2-3 and 5-8, and have removed the graphical abstract.
- Text is very small and unreadable in many cases
RESPONSE: All Figures have been remade in higher resolution, with larger label text, and the dimensions of all figures have been increased.
- Use of the font-size to show significance of the value is in general great idea; however, some fonts are not readable in figures 2 and 3.
RESPONSE: We understand and agree with the reviewer. All r values in Figures 2 and 3 have been enlarged to the same size.
- Figures 5 - 8 are almost impossible to read and thus they are not understandable
RESPONSE: Figures 5-8 have been rearranged and increased in size to make clear the full data distributions for each biomarker across HR+ and HER2+ cancers. An alternate approach would be to rearrange them into landscape on their own page, and we welcome comments on the preferred approach.
- Authors should optimize ranges in all X-Axis to fit the data; there is no need from large blank spaces in the plots
RESPONSE: We agree with the reviewer, and have increased the size of each plot and removed the legends from all but the top left plot in Figures 5-8. With this change, there is are no more obscured or partially obscured datapoints, so the range and skewing of the datasets becomes more visible. In each figure, the between-receptor status plots for each biomarker pair are standardized to each other in order to accommodate the full range of each dataset and make the visual comparisons valid between HR+ and Her2+ for each biomarker. For example, in Figure 5b, both the HR+ plot and HER2+ have datapoints between 150 and 200 on the x-axis, and the HER2+ datapoint is near the max value of the y-axis. This is true of the axis-formatting for all plots in Figures 5-8.
- The authors should show also parameters of the particular regression models for the trends plotted in the figures 5 - 8.
RESPONSE: The slopes, confidence intervals, and intercepts are now all shown in Tables S3 and S4.
Reviewer 2 Report
Comments and Suggestions for Authors
Aim
The Authors conducted a systematic longitudinal analysis of CTCs isolated from 51 patients with metastatic breast cancer during the course of their treatment to deepen our understanding of CTC contributions to breast cancer progression. The general goal is too vague. The Authors should detail the aim of the study and explain which clinical and biological endpoints were set.
Introduction
- “in an unbiased way”: please explain the meaning of this sentence, taking into consideration that although there is no selection of CTCs by Rarecyte, target cells are recognized by using immunostaining for a panel of proteins, which could be downregulated or expressed by non-tumor cells.
Materials and Methods
- Please define the case series indicating a) at which time points blood samples were collected, if there is a baseline blood samples, i.e. collected before the administration of anticancer therapies, and b) which therapeutic protocols were administered.
- Please indicate the antibody clone, isotype, host.
- Please indicate which CD45+ target cells do you mean to count to the purpose of this work, if they are included in CTC clusters (heterotypic clusters) or coexpress some epithelial marker etc.
- Please explain the difference between full and reduced CTC models, and indicate the number of cases per each dataset analyzed, and the reason why T2CTC size is smaller.
Results
- Figure 1: please explain the meaning of ‘clustering behaviour’
- Please report how treatment might have influenced CTC count and CTC biomarker composition if longitudinal analysis was performed
- Please indicate which time point for blood collection and CTC analysis was used to perform correlation analysis
- Table 4: please explain which kind of statistics was used.
- Please provide further clarification about the claim “EpCAM and Trop2 have overlapping but distinct roles in CTC shedding, clustering, and homing to distant metastatic sites” and explain which are the results that support and how did you come to that conclusion.
- “While the smaller size of the T2CTC dataset relative to the cCTC dataset does warrant caution, the results suggest that shedding of Trop2+ CTCs may be a feature of bone metastatic disease.”: as bone metastases are more frequent in HR+ breast cancer, please verify if there is enrichment for a specific subtype within the group of Trop2+ cases.
- “Additionally, larger clusters (those with greater than 2 cells) differed both at baseline and by slope between receptor subtypes after diagnosis with brain metastasis, suggesting that larger cluster size corresponds more to brain metastasis in HER2+ than in HR+ disease.” Please clarify this concept and which data support this conclusion.
- “Altogether, our results illustrate the critical roles of EpCAM and Trop2 in CTC clustering behavior after metastatic diagnosis, giving clarity to a clinical literature largely confounded by inter- and intra-patient heterogeneity in CTC shedding and clustering behavior”. Not clear, please reformulate.
- Please explain if data will have clinical implications and how they could be used.
Conclusions
- Please list the main conclusions supported by data.
- “our work is an important first step in unravelling the mechanisms underlying successful metastasis”. I suggest to reformulate as there is no experimental data that support the biological meaning of the Authors’observations.
Comments on the Quality of English Language
English must be improved, please see not appropriate use of 'between' instead of 'among' when your refer to breast cancer subtypes, and several other sentences not clear to the reader
Author Response
Reviewer #2
… to deepen our understanding of CTC contributions to breast cancer progression. The general goal is too vague. The Authors should detail the aim of the study and explain which clinical and biological endpoints were set.
RESPONSE: We thank the reviewer for his/her useful comments. We have substantially expanded and revised the relevant sections, as detailed below. We have addressed the aim of the study in lines 150-153, and addressed endpoints in the heavily revised section 2.4 (lines 210-244).
Introduction
- “in an unbiased way”: please explain the meaning of this sentence, taking into consideration that although there is no selection of CTCs by Rarecyte, target cells are recognized by using immunostaining for a panel of proteins, which could be downregulated or expressed by non-tumor cells.
RESPONSE: We agree with the reviewer and have removed this language from the revised manuscript.
Materials and Methods
- Please define the case series indicating a) at which time points blood samples were collected, if there is a baseline blood samples, i.e. collected before the administration of anticancer therapies, and b) which therapeutic protocols were administered.
RESPONSE: We have clarified that due to rolling enrollment periods, the statistical baseline in our analysis is the date of diagnosed metastasis rather than a baseline blood sample. This is discussed at length in lines 210-244.
- Please indicate the antibody clone, isotype, host.
RESPONSE: We have addressed these requirements. Please see lines 179-182 of the revised manuscript.
- Please indicate which CD45+ target cells do you mean to count to the purpose of this work, if they are included in CTC clusters (heterotypic clusters) or coexpress some epithelial marker etc.
RESPONSE: We have added text clarifying that only CD45+ cells negative for all epithelial markers were included. Please see lines 194-198 of the revised manuscript.
- Please explain the difference between full and reduced CTC models, and indicate the number of cases per each dataset analyzed, and the reason why T2CTC size is smaller.
RESPONSE: We thank the Reviewer for his/her comments. We have substantially expanded Section 2.5, and have addressed that the cases per each dataset analyzed are those listed in Tables 1 and 2. Please see lines 157-161 and 273-277. The full and reduced models are explained in far greater detail, please see lines 260-277. We have added new text to address the smaller size of the T2CTC dataset in lines 190-193 of the revised manuscript.
Results
- Figure 1: please explain the meaning of ‘clustering behaviour’
RESPONSE: We thank the Reviewer for his/her comment. We have removed the phrase, and now all instances refer simply to “CTC clustering.”
- Please report how treatment might have influenced CTC count and CTC biomarker composition if longitudinal analysis was performed
RESPONSE: We thank the Reviewer for his/her comment. We have added text to address to possible effects of treatment. Please see lines 238-245 of the revised manuscript.
- Please indicate which time point for blood collection and CTC analysis was used to perform correlation analysis
RESPONSE: We thank the Reviewer for his/her comment. We have now added further text to clarify that correlation analysis was performed on averaged biomarker values and therefore represent multiple timepoints. Please see lines 313-315 and 460-464 of the revised manuscript.
- Table 4: please explain which kind of statistics was used.
RESPONSE: We have revised the Table title to specify that the results detailed in Tables 3 and 4 are LMM results, and have expanded the text section referencing these tables to add additional clarity. Please see lines 327-329 and 339 of the revised manuscript.
- Please provide further clarification about the claim “EpCAM and Trop2 have overlapping but distinct roles in CTC shedding, clustering, and homing to distant metastatic sites” and explain which are the results that support and how did you come to that conclusion.
RESPONSE: We thank the Reviewer for his/her comments. We have now added citations to the references that lay the foundation for this assertion and added references to the relevant figures (Figure 5b and 5d and Table S1). Please see lines 369-375 of the revised manuscript.
- “While the smaller size of the T2CTC dataset relative to the cCTC dataset does warrant caution, the results suggest that shedding of Trop2+ CTCs may be a feature of bone metastatic disease.”: as bone metastases are more frequent in HR+ breast cancer, please verify if there is enrichment for a specific subtype within the group of Trop2+ cases.
RESPONSE: We have added references to the relevant figures and clarified our results with regard to enrichment of Trop2 cases by subtype (Figure 4d, 8d, and Table S2). Please see lines 411-414 of the revised manuscript.
- Additionally, larger clusters (those with greater than 2 cells) differed both at baseline and by slope between receptor subtypes after diagnosis with brain metastasis, suggesting that larger cluster size corresponds more to brain metastasis in HER2+ than in HR+ disease.” Please clarify this concept and which data support this conclusion.
RESPONSE: We thank the Reviewer for his/her comments. We have now specified and added references to Figure 8b and Table S2, please see line 416 of the revised manuscript.
- Altogether, our results illustrate the critical roles of EpCAM and Trop2 in CTC clustering behavior after metastatic diagnosis, giving clarity to a clinical literature largely confounded by inter- and intra-patient heterogeneity in CTC shedding and clustering behavior”. Not clear, please reformulate.
RESPONSE: We have reformulated this sentence for added clarity.
- Please explain if data will have clinical implications and how they could be used.
RESPONSE: We thank the Reviewer for his/her comment. We have now added text and discuss the clinical implications in lines 417-425 of the revised manuscript.
Conclusions
- Please list the main conclusions supported by data.
RESPONSE: We thank the Reviewer for his/her comment. We have now listed the main conclusions of our study in the revised conclusion section, please see lines 502-511 of the revised manuscript.
- “our work is an important first step in unravelling the mechanisms underlying successful metastasis”. I suggest to reformulate as there is no experimental data that support the biological meaning of the data.
RESPONSE: We thank the Reviewer for his/her suggestion. We have now revised this wording, please see lines 501-502 of the revised manuscript.
Comments on the Quality of English Language
English must be improved, please see not appropriate use of 'between' instead of 'among' when your refer to breast cancer subtypes, and several other sentences not clear to the reader
RESPONSE: We thank the reviewer for his/her comment. Specifically, we have used the phrase “between receptor subtypes”, in the context it is referring to the two receptor statuses included in the longitudinal analysis, HR+ and HER2+, i.e. “between the two receptor subtypes.”, eg, line 332. For any other instances, we have changed sentences accordingly. For example, lines 24 and 41 have been changed to “among,” as in context they are referring to all 3 subtypes. Further, line 294 has been changed to read “among.”, per reviewer’s request.
Round 2
Reviewer 1 Report
Comments and Suggestions for Authors
The revised version of the manuscript reflects my comments and authors make adequate changes. The authors considerably improved both, the scientific part as well as a style of the text in the revised version of the manuscript. I have no further comments.
Reviewer 2 Report
Comments and Suggestions for Authors.